# An End-to-End Model for Question Answering over Knowledge Base with Cross-Attention Combining Global Knowledge

## Abstract

With the rapid growth of knowledge bases (KBs) on the web, how to take full advantage of them becomes increasingly important. Knowledge base-based question answering (KB-QA) is one of the most promising approaches to access the substantial knowledge. Meantime, as the neural network-based (NN-based) methods develop, NN-based KB-QA has already achieved impressive results. However, previous work did not put emphasis on question representation, and the question is converted into a fixed vector regardless of its candidate answers. This simple representation strategy is unable to express the proper information of the question. Hence, we present an end-to-end neural network model to represent the questions and their corresponding scores dynamically according to the various candidate answer aspects via cross-attention mechanism. In addition, we leverage the global knowledge inside the underlying KB, aiming at integrating the rich KB information into the representation of the answers. Moreover, it also alleviates the out-of-vocabulary (OOV) problem, which helps the cross-attention model to represent the question more precisely. The experimental results on WebQuestions demonstrate the effectiveness of the proposed approach.

## 1 Introduction

As the amount of the knowledge bases (KBs) grows, people are paying more attention to seeking effective methods for accessing these precious intellectual resources. There are several tailor-made languages designed for querying KBs, such as SPARQL (Prudhommeaux and Seaborne, 2008). However, to handle such query languages, users are required to not only be familiar with the particular language grammars, but also be aware of the architectures of the KBs. By contrast, knowledge base-based question answering (KB-QA) (Unger et al., 2014), which takes natural language as query language, is a more user-friendly solution, and has become a research focus in recent years.

Given natural language questions, the goal of KB-QA is to automatically return answers from the KB. There are two mainstream research directions for this task: semantic parsing-based (SP-based) (Zettlemoyer and Collins, 2009, 2012; Kwiatkowski et al., 2013; Cai and Yates, 2013; Berant et al., 2013; Yih et al., 2015, 2016; Reddy et al., 2016) and information retrieval-based (IR-based) (Yao and Van Durme, 2014; Bordes et al., 2014a,b, 2015; Dong et al., 2015; Xu et al., 2016a,b) methods. SP-based methods usually focus on constructing a semantic parser that could convert natural language questions into structured expressions like logical forms. IR-based methods usually search answers from the KB based on the information conveyed in questions, where ranking techniques are often adopted to make correct selections from candidate answers.

Recently, with the progress of deep learning, neural network-based (NN-based) methods have been introduced to the KB-QA task (Bordes et al., 2014b). They belong to IR-based methods. Different from previous methods, NN-based methods represent both of the questions and the answers as semantic vectors. Then the complex process of KB-QA could be converted into a similarity matching process between an input question and its candidate answers in a semantic space. The candidates with the highest similarity score will be selected as the final answers. Because they are adaptive and robust, NN-based methods have at-

tracted more and more attention, and this paper also focuses on using end-to-end neural networks to answer questions over knowledge base.

In NN-based methods, the crucial step is to compute the similarity score between a question and a candidate answer, where the key is to learn their representations. Previous methods put more emphasis on learning representation of the answer end. For example, Bordes et al. (2014a) consider the importance of the subgraph of the candidate answer. Dong et al. (2015) make use of the context and the type of the answer. However, the representation of the question end is oligotrophic. Existing approaches often represent a question into a single vector using simple bag-of-words (BOW) model (Bordes et al., 2014a,b), whereas the relatedness to the answer end is neglected. We argue that a question should be represented differently according to the different focuses of various answer aspects[1].

Take the question "Who is the president of France?" and one of its candidate answers "Francois Hollande" as an example. When dealing with the answer entity `Francois Holland`, "president" and "France" in the question is more focused, and the question representation should bias towards the two words. While facing the answer type `/business/board_member`, "Who" should be the most prominent word. Meantime, some questions may value answer type more than other answer aspects. While in some other questions, answer relation may be the most important information we should consider, which is dynamic and flexible corresponding to different questions and answers. Obviously, this is an attention mechanism, which reveals the mutual influences between the representation of questions and the corresponding answer aspects.

We believe that such kind of representation is more expressive. Dong et al. (2015) represents questions using three CNNs with different parameters when dealing with different answer aspects including answer path, answer context and answer type. We argue that simply selecting three independent CNNs is mechanical and inflexible. Thus, we go one step further, and propose a cross-attention based neural network to perform KB-QA. The cross-attention model, which stands for the mutual attention between the question and the answer aspects, contains two parts: the answer-towards-question attention part and the question-towards-answer attention part. The former help learn flexible and adequate question representation, and the latter help adjust the question-answer weight, getting the final score. We illustrate in section 3.2 for more details. In this way, we formulate the cross-attention mechanism to model the question answering procedure. Note that our proposed model is an entire end-to-end approach which only depends on training data. Some integrated systems which use extra patterns and resources are not directly comparable to ours. Our target is to explore a better solution following the end-to-end KB-QA technical path.

Moreover, we notice that the representations of the KB resources (entities and relations) are also limited in previous work. specifically, they are often learned barely on the QA training data, which results in two limitations. 1) The global information of the KB is deficient. For example, if question-answer pair (q, a) appears in the training data, and the global KB information implies us that $a'$ is similar to a[2], denoted by (a $\sim a'$), then (q, $a'$) is more probable to be right. However, current QA training mechanism cannot guarantee (a $\sim a'$) could be learned. 2) The problem of out-of-vocabulary (OOV) stands out. Due to the limited coverage of the training data, the OOV problem is common while testing, and many answer entities in testing candidate set have never been seen before. The attention of these resources become the same because they shared the same OOV embedding, and this will do harm to the proposed attention model. To tackle these two problems, we additionally incorporates KB itself as training data for training embeddings besides original question-answer pairs. In this way, the global structure of the whole knowledge could be captured, and the OOV problem could be alleviated naturally.

In summary, the contributions are as follows.
1) We present a novel cross-attention based NN model tailored to KB-QA task, which considers the mutual influence between the representation of questions and the corresponding answer aspects.
2) We leverage the global KB information, aiming at represent the answers more precisely. It also alleviates the OOV problem, which is very helpful to the cross-attention model.

---

[1] An answer aspect could be the answer entity itself, the answer type, the answer context, etc.

[2] The complete KB is able to offer this kind of information, e.g., a and $a'$ share massive context.

3) The experimental results on the open dataset WebQuestions demonstrate the effectiveness of the proposed approach.

## 2 Overview

The goal of KB-QA task could be formulated as follows. Given a natural language question $q$, the system returns an entity set $A$ as answers. The architecture of our proposed KB-QA system is shown in Figure 1, which illustrates the basic flow of our approach. First, we identify the topic entity of the question, and generate candidate answers from Freebase. Then, a cross-attention based neural network is employed to represent the question under the influence of the candidate answer aspects. Finally, the similarity score between the question and each corresponding candidate answer is calculated, and the candidates with highest score will be selected as the final answers[3].

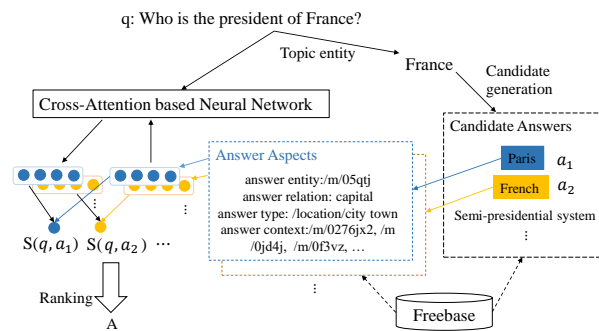

Figure 1: The overview of the proposed KB-QA system.

We utilize Freebase (Bollacker et al., 2008) as our knowledge base. It has more than 3 billion facts, and is used as the supporting KB for many QA tasks. In Freebase, the facts are represented by subject-predicate-object triples $(s, p, o)$. For clarity, we call each basic element a resource, which could be either an entity or a relation. For example, (/m/0f8l9c, location.country.capital,/m/05qtj)[4] describes the fact that the capital of France is Paris, where /m/0f8l9c and /m/05qtj are entities denoting France and Paris respectively, and location.country.capital is a relation.

---

[3]We also adopt a margin strategy to obtain multiple answers for a question and this will be explained in the next section.

[4]Note that the Freebase prefixes are omitted for neatness.

## 3 Our Approach

### 3.1 Candidate Generation

All the entities in Freebase should be candidate answers ideally, but in practice, this is time consuming and not really necessary. For each question $q$, we use Freebase API (Bollacker et al., 2008) to identify a topic entity, which could be simply understood as the main entity of the question. For example, France is the topic entity of question "Who is the president of France?". Freebase API method is able to resolve as many as 86% questions if we use the top1 result (Yao and Van Durme, 2014). After getting the topic entity, we collect all the entities directly connected to it and the ones connected with 2-hop[5]. These entities constitute a candidate set $C_q$.

### 3.2 The Neural Cross-Attention Model

We present a cross-attention based neural network, which represents the question dynamically according to different answer aspects, also considering their connections. Concretely, each aspect of the answer focuses on different words of the question and thus decides how the question is represented. Then the question pays different attention to each answer aspect to decide their weights. Figure 2 is the architecture of our model. We will illustrate how the system works as follows.

#### 3.2.1 Question Representation

First of all, we have to obtain the representation of each word in the question. These representations retain all the information of the question, and could serve the following steps. Suppose question $q$ is expressed as $q = (x_1, x_2, ..., x_n)$, where $x_i$ denotes the $i$th word. As shown in Figure 2, we first look up a word embedding matrix $E_w \in \mathbb{R}^{d \times v_w}$ to get the word embeddings, which is randomly initialized, and updated during the training process. Here, $d$ means the dimension of the embeddings and $v_w$ denotes the vocabulary size of natural language words.

Then, the embeddings are fed into a long short-term memory (LSTM) (Hochreiter and Schmidhuber, 1997) networks. LSTM has been proven to be effective in many natural language processing (NLP) tasks such as machine translation (Sutskever et al., 2014) and dependency parsing (Dyer et al., 2015), and it is adept in harnessing long

---

[5]For example, (/m/0f8l9c, governing_officials, government.position_held.office_holder, /m/02qg4z) is a 2-top connection.

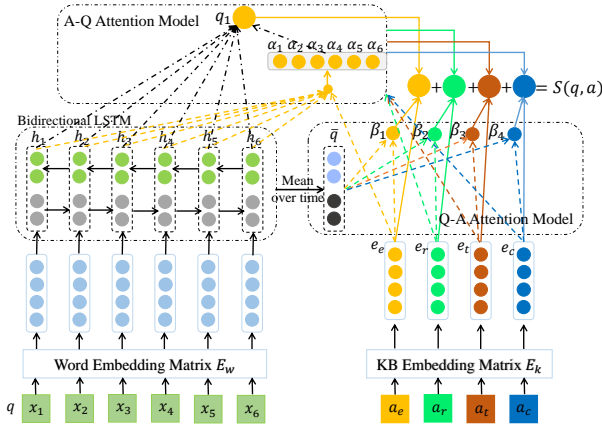

Figure 2: The architecture of the proposed attention-based neural network. Note that only one aspect(in orange color) is depicted for clarity. The other three aspects follow the same way.

sentences. Note that if we use unidirectional L-STM, the outcome of a specific word contains only the information of the words before it, whereas the words after it are not taken into account. To avoid this, we employ bidirectional LSTM as Bahdanau (2015) does, which consists of both forward and backward networks. The forward LSTM handles the question from left to right, and the backward LSTM processes in the reverse order. Thus, we could acquire two hidden state sequences, one from the forward one $(\overrightarrow{h_1}, \overrightarrow{h_2}, ..., \overrightarrow{h_n})$ and the other from the backward one $(\overleftarrow{h_1}, \overleftarrow{h_2}, ..., \overleftarrow{h_n})$. We concatenate the forward hidden state and the backward hidden state of each word, resulting in $h_j = [\overrightarrow{h_j}; \overleftarrow{h_j}]$. The hidden unit of forward and backward LSTM is $\frac{d}{2}$, so the concatenated vector is of dimension $d$. In this way, we obtain the representation of each word in the question.

### 3.2.2 Answer aspect representation
We directly use the embedding for each answer aspect through the KB embedding matrix $E_k \in \mathbb{R}^{d \times v_k}$. Here, $v_k$ means the vocabulary size of the KB resources. The embedding matrix is randomly initialized and learned during training, and could be further enhanced with the help of global information as described in Section 3.3. Concretely, we employ four kinds of answer aspects: answer entity $a_e$, answer relation $a_r$, answer type $a_t$ and answer context $a_c$. Their embeddings are denoted as $e_e$, $e_r$, $e_t$ and $e_c$, respectively. It is worth noting that the answer context consists of multiple KB resources, and we denote it as $(c_1, c_2, ..., c_m)$. We first acquire their KB embeddings $(e_{c_1}, e_{c_2}, ..., e_{c_m})$ through $E_k$, then calculate

an average embedding by $e_c = \frac{1}{m} \sum_{i=1}^{m} e_{c_i}$.

### 3.2.3 Cross-Attention model
The most crucial part of the proposed approach is the cross-attention mechanism. The cross-attention mechanism is composed of two parts: the answer-towards-question attention part and the question-towards-answer attention part.

**• Answer-towards-question(A-Q) attention**

Based on our assumption, each answer aspect should focus on different words of the same question. The extent of attention can be measured by the relatedness between each word representation $h_j$ and an answer aspect embedding $e_i$. We propose the following formulas to calculate the weights.

$$\alpha_{ij} = \frac{\exp(\omega_{ij})}{\sum\limits_{k=1}^{n} \exp(\omega_{ik})} \quad (1)$$

$$\omega_{ij} = f(W^T[h_j; e_i] + b) \quad (2)$$

Here, $\alpha_{ij}$ denotes the weight of attention from answer aspect $e_i$ to the $j$th word in the question, where $e_i \in \{e_e, e_r, e_t, e_c\}$. $f(\cdot)$ is a non-linear activation function, such as hyperbolic tangent transformation here. Let $n$ be the length of the question. $W \in \mathbb{R}^{2d \times d}$ is an intermediate matrix and $b$ is the offset. Both of them are randomly initialized and updated during training. Subsequently, according to the specific answer aspect $e_i$, the attention weights are employed to calculate a weighted sum of the hidden representations, resulting in a semantic vector that represent the question.

$$q_i = \sum_{j=1}^{n} \alpha_{ij} h_j \quad (3)$$

The similarity score of the question $q$ and this particular candidate answer aspect $e_i$ ($e_i \in \{e_e, e_r, e_t, e_c\}$) could be defined as follows.

$$S(q, e_i) = h(q_i, e_i) \quad (4)$$

The scoring function $h(\cdot)$ is computed as the inner product between the sentence representation $q_i$, which has already carried the attention from the answer aspect part, and the corresponding answer aspect $e_i$, and is parametrized into the network and updated during the training process.

**• Question-towards-answer(Q-A) attention**

Intuitively, different question should value the four answer aspect differently. Since we have already calculated the scores of $(q, e_i)$, we define the final similarity score of the question $q$ and each candidate answer $a$ as follows.

$$S(q, a) = \sum_{e_i \in \{e_e, e_r, e_t, e_c\}} \beta_{e_i} \cdot S(q, e_i) \quad (5)$$

$$\beta_{e_i} = \frac{\exp(\omega_{e_i})}{\sum_{e_k \in \{e_e, e_r, e_t, e_c\}} \exp(\omega_{e_k})} \quad (6)$$

$$\omega_{e_i} = f\left(W^T \cdot [\overline{q}; e_i] + b\right) \quad (7)$$

$$\overline{q} = \frac{1}{n} \sum_j^n h_j \quad (8)$$

Here $\beta_{e_i}$ denotes the attention of question towards answer aspects, indicating which answer aspect should be more focused in one $(q, a)$ pair. $W \in \mathbb{R}^{2d \times d}$ is also a intermediate matrix as in the answer-towards-question attention part, and $b$ is an offset value.[6] $\overline{q}$ is calculated by averagely pooling all the bi-directional LSTM hidden state sequences, resulting a vector which represents the question to determine which answer aspect should be more focused.

The proposed cross-attention model could also be intuitively interpreted as a re-reading mechanism (Hermann et al., 2015). Our aim is to select correct answers from a candidate set. When we judge a candidate answer, suppose we first look at its type, and we will re-read the question to find out which part of the question should be more focused (handling attention). Then we go to next aspect and re-read the question again, until the all the aspects are utilized. After we read all the answer aspects and get all the scores, the final similarity score between question and answer should be a weighted sum of all the scores. We believe that this mechanism is beneficial for the system to better understand the question with the help of the answer aspects, and it may lead to a performance promotion.

### 3.2.4 Training

We first construct the training data. Since we have $(q, a)$ pairs as supervision data, candidate set $C_q$ can be divided into two subsets, namely, correct answer set $P_q$ and wrong answer set $N_q$. For each correct answer $a \in P_q$, we randomly select $k$ wrong answers $a' \in N_q$ as negative examples. For some topic entities, there may be not enough wrong answers to acquire $k$ wrong answers. Under this circumstance, we extend $N_q$ from other randomly selected candidate set $C_q'$. With the generated training data, we are able to make use of pairwise training. The training loss is given as follows, which is a hinge loss.

$$L_{q,a,a'} = [\gamma + S(q, a') - S(q, a)]_+ \quad (9)$$

---

[6]Note that the $W$ and $b$ in the two attention part is different and independent.

where $\gamma$ is a positive real number that ensure a margin between positive and negative examples. And $[z]_+$ means $max(0, z)$. The intuition of this training strategy is to guarantee the score of positive question-answer pairs be higher than negative ones with a margin. The objective function is as follows.

$$\min \sum_q \frac{1}{|P_q|} \sum_{a \in P_q} \sum_{a' \in N_q} L_{q,a,a'} \quad (10)$$

We adopt stochastic gradient descent (SGD) to implement the learning process, shuffled mini-batches are utilized.

### 3.2.5 Inference

In testing stage, given the candidate answer set $C_q$, we have to calculate $S(q, a)$ for each $a \in C_q$, and find out the maximum value $S_{max}$.

$$S_{\max} = \arg\max_{a \in C_q}\{S(q, a)\} \quad (11)$$

It is worth noting that many questions have more than one answer, so it is improper to set the candidate answer which have the maximum value as the final answer. Instead, we take advantage of the margin $\gamma$. If the score of an candidate answer is within the margin compared with $S_{max}$, we put it in the final answer set.

$$A = \{\hat{a}|S_{\max} - S(q, \hat{a}) < \gamma\} \quad (12)$$

### 3.3 Combining Global Knowledge

In this section, we elaborate how the global information of a KB could be leveraged. As stated before, we try to take into account the complete knowledge information of the KB. To this end, we adopt TransE model (Bordes et al., 2013) and integrate its outcome into our training process. In TransE, relations are considered as translations in the embedding space. For consistency, we denote each fact as $(s, p, o)$. TransE utilizes pairwise training strategy as well. Randomly sampled corrupted facts $(s', p, o')$ are the negative examples. The distance measure $d(s + p, o)$ is defined as $\|s + p - o\|_2^2$. And the training loss is given as follows.

$$L_k = \sum_{(s,p,o) \in S} \sum_{(s',p,o') \in S'} [\gamma_k + d(s + p, o) - d(s' + p, o')]_+ \quad (13)$$

Where $S$ is the set of KB facts and $S'$ is the corrupted facts. In our QA task, we filter out the completely unrelated facts to save time. Specifically, we first collect all the topic entities of all the questions as initial set. Then, we expand the set by adding directly connected and 2-hop entities. Finally, all the facts containing these entities form

the positive set, and the negative facts are randomly corrupted. This is a compromising solution due to the large scale of Freebase. To employ the global information in our training process, we adopt a multi-task training strategy. Specifically, we perform KB-QA training and TransE training in turn. After each epoch of KB-QA training, 100 epochs of TransE training are conducted, and the embeddings of the KB resources are shared and updated during both training processes. The proposed training process ensures that the global KB information act as additional supervision, and the interconnections among the resources are fully considered. In addition, as more KB resources are involved, the OOV problem is relieved. Since all the OOV resources have exactly the same attention towards a question, it will weaken the effectiveness of the attention model. So the alleviation of OOV is able to bring additional benefits to the attention model.

## 4 Experiments

To evaluate the proposed method, we conduct experiments on WebQuestions (Berant et al., 2013) dataset that includes 3,778 question-answer pairs for training and 2,032 for testing. The questions are collected from Google Suggest API, and the answers are labeled manually by Amazon MTurk. All the answers are from Freebase. We use three-quarter of the training data as training set, and the left as validate set. We use $F_1$ score as evaluation matric, and the average result is computed by the script provided by Berant et al. (2013).

Note that our proposed approach is an entire end-to-end method, which totally depends on training data. It is worth noting that Yih et al. (2015; 2016) achieves much higher $F_1$ scores than other methods. Their staged system is able to address more questions with constraints and aggregations. However, their approach applies numbers of manually designed rules and features, which come from the observations on the training set questions. These particular manual efforts reduce the adaptability of their approach. Moreover, there are some integrated systems such as Xu et al. (2016a; 2016b) achieve higher $F_1$ scores which leverage Wikipedia free text as external knowledge, so their systems are not directly comparable to ours.

### 4.1 Settings

For KB-QA training, we use mini-batch stochastic gradient descent to minimize the pairwise training loss. The minibatch size is set to 100. The learning rate is set to 0.01. Both the word embedding matrix $E_w$ and KB embedding matrix $E_v$ are normalized after each epoch. The embedding size d = 512, then the hidden unit size is 256. Margin $\gamma$ is set to 0.6. Negative example number k = 2000. We set the embedding dimension to 512 in TransE training process, and the minibatch size is also 100. $\gamma_k$ is set to 1. All these hyperparameters of the proposed network is determined according to the performance on the validate set.

### 4.2 Results

**The effectiveness of the proposed approach**

To demonstrate the effectiveness of the proposed approach, we compare our method with state-of-the-art end-to-end NN-based methods.

| Methods | $F_1$ |
|---|---|
| Bordes et al., 2014b | 29.7 |
| Bordes et al., 2014a | 39.2 |
| Yang et al., 2014 | 41.3 |
| Dong et al., 2015 | 40.8 |
| Bordes et al., 2015 | 42.2 |
| **our approach** | **42.9** |

Table 1: The evaluation results on WebQuestions.

Table 1 shows the results on WebQuestions dataset. Bordes et al. (2014b) apply BOW method to obtain a single vector for both questions and answers. Bordes et al. (2014a) further improve their work by proposing the concept of subgraph embeddings. Besides the answer path, the subgraph contains all the entities and relations connected to the answer entity. The final vector is also obtained by bag-of-words strategy. Yang et al. (2014) follow the SP-based manner, but uses embeddings to map entities and relations into KB resources, then the question can be converted into logical forms. They jointly consider the two mapping processes. Dong et al. (2015) use three columns of C-NNs to represent questions corresponding to three aspects of the answers, namely the answer context, the answer path and the answer type. Bordes et al. (2015) put KB-QA into the memory networks framework (Sukhbaatar et al., 2015), and achieves the state-of-the-art performance of end-to-end methods. Our approach employs bidirectional LSTM, cross-attention model and global KB information.

From the results, we observe that our approach achieves the best performance of all the end-to-end

methods on WebQuestions. Bordes et al. (2014b; 2014a; 2015) all utilize BOW model to represent the questions, while ours takes advantage of the attention of answer aspects to dynamically represent the questions. Also note that Bordes et al. (2015) uses additional training data such as Reverb (Fader et al., 2011) and their original dataset Simple-Questions. Dong et al. (2015) employs three fixed CNNs to represent questions, while ours is able to express the focus of each unique answer aspect to the words in the question. Besides, our approach employs the global KB information. So, we believe that the results faithfully shows that the proposed approach is more effective than the other competitive methods.

**Model Analysis**

In this part, we further discuss the impacts of the components of our model. Table 2 indicates the effectiveness of different parts in the model.

| Methods | $F_1$ |
|---|---|
| LSTM | 38.2 |
| Bi_LSTM | 39.1 |
| Bi_LSTM+A-Q-ATT | 41.6 |
| Bi_LSTM+C-ATT | 41.8 |
| Bi_LSTM+GKI | 40.4 |
| Bi_LSTM+A-Q-ATT+GKI | 42.6 |
| Bi_LSTM+C-ATT+GKI | 42.9 |

Table 2: The ablation results of our models.

*LSTM* employs unidirectional LSTM, and uses the last hidden state as the question representation. *Bi_LSTM* adopts a bidirectional LSTM. *A-Q-ATT* denotes the answer-towards-question attention part, and C-ATT stands for our cross-attention. *GKI* means global knowledge information. *Bi_LSTMS+C-ATT+GKI* is our full proposed approach. From the results, we could observe the following.

1) *Bi_LSTM+C-ATT* dramatically improves the $F_1$ score by 2.7 points compared with *Bi_LSTM*, 0.2 points higher than *Bi_LSTM+A-Q-ATT*. Similarly, *Bi_LSTM+C-ATT+GKI* significantly outperforms *Bi_LSTM+GKI* by 2.5 points, improving *Bi_LSTM+A-Q-ATT+GKI* by 0.3 points. The results prove that the proposed cross-attention model is effective.

2) *Bi_LSTM+GKI* performs better than *Bi_LSTM*, and achieves an improvement of 1.3 points. Similarly, *Bi_LSTM+C-ATT+GKI* improves *Bi_LSTM+C-ATT* by 1.1 points, which indicates that the proposed training strategy

successfully leverages the global information of the underlying KB.

3) *Bi_LSTM+C-ATT+GKI* achieves the best performance as we expected, and improves the original *Bi_LSTM* dramatically by 3.8 points. This directly shows the power of the attention model and the global KB information.

To illustrate the effectiveness of the attention mechanism clearly, we present the attention weights of a question in the form of heat map as shown in Figure 3.

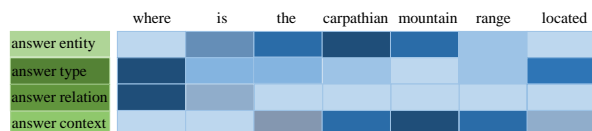

Figure 3: The visualized attention heat map. Answer entity: /m/06npd(Slovakia), answer relation: partially_containedby, answer type: /location/country, answer context: (/m/04dq9kf, /m/01mp, ...)

From this example we observe that our methods is able to capture the attention properly. It is instructive to figure out the attention part of the question when dealing with different answer aspects. The heat map will help us understand which parts are most useful for selecting correct answers. For instance, from Figure 3, we can see that location.country is paying great attention to "Where", indicating that "Where" is much more important than the other parts in the question when dealing with this type. In other words, the other parts are not that crucial since "Where" is strongly implying that the question is asking about a location. As for Q-A attention part, we see that answer type and answer relation are more important than other answer aspects in this example.

### 4.3 Error Analysis

We randomly sample 100 imperfectly answered questions and categorize the errors into two main classes as follows.

**Wrong attention**

In some occasions (17 in 100 questions, 17%), we find the generated attention weights unreasonable. For instance, for question "What are the songs that Justin Bieber wrote?", answer type /music/composition pays the most attention on "What" rather than "songs". We think this is due to the bias of the training data, and we believe these errors could be solved by introducing more instructive training data.

**Complex questions and label errors**

Another challenging problem is the complex questions (34%). For example, "When was the last time Knicks won the championship?" is actually to ask the last championship, but the predicted answers give all the championships. This is due to that the model cannot learn what "last" mean in the training process. In addition, the label mistakes also influence the evaluation (3%), such as, "What college did John Nash teach at?", where the labeled answer is `Princeton University`, but `Massachusetts Institute of Technology` should also be an answer, and the proposed method is able to answer it correctly. Other errors include topic entity generation error and the multiple answers error (giving more answers than expected). We guess these errors are caused by the simple implementations of the related steps in our method, and we will not explain them in detail due to space limitation.

## 5 Related Work

### 5.1 Neural Network-based KB-QA

In recent years, deep neural networks have been applied to many NLP tasks, showing promising results. Bordes et al. (2014b) was the first to introduce NN-based method to solve KB-QA problem. The questions and KB triples were represented by vectors in a low dimensional space. Thus the cosine similarity could be used to find the most possible answer. BOW method was employed to obtain a single vector for both the questions and the answers. Pairwise training was utilized, and the negative examples were randomly selected from the KB facts. Bordes et al. (2014a) further improved their work by proposing the concept of subgraph embeddings. The key idea was to involve as much information as possible in the answer end. Besides the answer triple, the subgraph contained all the entities and relations connected to the answer entity. The final vector was also obtained by bag-of-words strategy.

Yih et al. (2014) focused on single-relation questions. The KB-QA task was divided into two steps. Firstly, they found the topic entity of the question. Then, the rest of the question was represented by CNNs and used to match relations. Yang et al. (2014) tackled entity and relation mapping as joint procedures. Actually, these two methods followed the SP-based manner, but they took advantage of neural networks to obtain intermediate mapping results.

The most similar work to ours is Dong et al. (2015). They considered the different aspects of answers, using three columns of CNNs to represent questions respectively. The difference is that our approach uses cross-attention mechanism for each unique answer aspect, so the question representation is not fixed to only three types. Moreover, we utilize the global KB information.

Xu et al. (2016a; 2016b) proposed integrated systems to address KB-QA problems incorporating Wikipedia free text, in which they used multichannel CNNs to extract relations.

### 5.2 Attention-based Model

The attention mechanism has been widely used in different areas. Bahdanau et al. (2015) first applied attention model in NLP. They improved the encoder-decoder Neural Machine Translation (NMT) framework by jointly learning align and translation. They argued that representing source sentence by a fixed vector is unreasonable, and proposed a soft-align method, which could be understood as attention mechanism. Rush et al. (2015) implemented sentence-level summarization task. They utilized local attention-based model that generated each word of the summary conditioned on the input sentence.

Yin et al. (2016) tackled simple question answering by an attentive convolutional neural network. They stacked an attentive maxpooling above convolution layer to model the relationship between predicates and question patterns. Our approach differs from previous work in that we use attentions to help represent question dynamically, not generating current word from vocabulary as before.

## 6 Conclusion

In this paper, we focus on KB-QA task. Firstly, we consider the impacts of the different answer aspects when representing the question, and propose a novel cross-attention model for KB-QA. Specifically, we employ the focus of the answer aspects to each question word and the attention weights of the question towards the answer aspects. This kind of dynamic representation is more precise and flexible. Secondly, we leverage the global KB information, which could take full advantage of the complete KB, and also alleviate the OOV problem for the attention model. The extensive experiments demonstrate that the proposed approach could achieve better performance compared with state-of-the-art end-to-end methods.

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
