# Peer review of "An End-to-End Model for Question Answering over Knowledge Base with Cross-Attention Combining Global Knowledge"

_ACL 2017 — decision unknown_

[Official Review · Reviewer 1 · rating 4 · confidence 4]
soundness 5 · originality 3 · clarity 3 · impact 3 · substance 3 · appropriateness 5 · meaningful comparison 5 · presentation format Oral Presentation

26: An End-to-End Model for Question Answering over Knowledge Base with
Cross-Attention Combining Global Knowledge

This paper presents an approach for factoid question answering over a knowledge
graph (Freebase), by using a neural model that attempts to learn a semantic
correlation/correspondence between various "aspects" of the candidate answer
(e.g., answer type, relation to question entity, answer semantic, etc.) and a
subset of words of the question. A separate correspondence component is learned
for each "aspect" of the candidate answers. The two key contributions of this
work are: (1) the creation of separate components to capture different aspects
of the candidate answer, rather than relying on a single semantic
representation, and (2) incorporating global context (from the KB) of the
candidate answers.

The most interesting aspect of this work, in my opinion, is the separation of
candidate answer representation into distinct aspects, which gives us (the
neural model developer) a little more control over guiding the NN models
towards information that would be more beneficial in its decision making. It
sort of harkens to the more traditional algorithms that rely on feature
engineering. But in this case the "feature engineering" (i.e., aspects) is more
subtle, and less onerous. I encourage the authors to continue refining this
system along these lines.

While the high-level idea is fairly clear to a reasonably informed reader, the
devil in the details would make it hard for some audience to immediately grasp
key insights from this work. Some parts of the paper could benefit from more
explanation... Specifically:

(1) Context aspect of candidate answers (e_c) is not clearly explained in the
paper. Therefore, the last two sentences of Section 3.2.2 seem unclear.

(2) Mention of OOV in the abstract and introduction need more explanation. As
such, I think the current exposition in the paper assumes a deep understanding
of prior work by the reader.

(3) The experiments conducted in this paper restrict comparison to IR-based
system -- and the reasoning behind this decision is reasonable. But it is not
clear then why the work of Yang et al. (2014) -- which is described to be
SP-based -- is part of the comparison. While, I am all for including more
systems in the comparison, there seem to be some inconsistencies in what should
and should not be compared. Additionally, I see not harm in also mentioning the
comparable performance numbers for the best SP-based systems.

I observe in the paper that the embeddings are learned entirely from the
training data. I wonder how much impact the random initialization of these
embeddings has on the end performance. It would be interesting to determine
(and list) the variance if any. Additionally, if we were to start with
pre-trained embeddings (e.g., from word2vec) instead of the randomly
initialized ones, would that have any impact?

As I read the paper, one possible direction of future work that occurred to me
was to possibly include structured queries (from SP-based methods) as part of
the cross-attention mechanism. In other words, in addition to using the various
aspects of the candidate answers as features, one could include structured
queries that generate the produce that candidate answer as an additional aspect
of the candidate answer. An attention mechanism could then also focus on
various parts of the structured query, and its (semantic) matches to the input
question as an additional signal for the NN model. Just a thought.

Some notes regarding the positioning of the paper:

I hesitate to call the model proposed here "attention" models, because (per my
admittedly limited understanding) attention mechanisms apply to
"encoder-decoder" situations, where semantics expressed in one structured form
(e.g., image, sentence in one language, natural language question, etc.) are
encoded into an abstract representation, and then generated into another
structured form (e.g., caption, sentence in another language, structured query,
etc.). The attention mechanism allows the "encoder" to jump around and attend
to different parts of the input (instead of sequentially) as the output is
being generated by the decoder. This paper does not appear to fit this notion,
and may be confusing to a broader audience.

------

Thank you for clarifications in the author response.

[Official Review · Reviewer 2 · rating 4 · confidence 4]
soundness 5 · originality 3 · clarity 4 · impact 3 · substance 4 · appropriateness 5 · meaningful comparison 5 · presentation format Oral Presentation

- Strengths:
This paper contributes to the field of knowledge base-based question answering
(KB-QA), which is to tackle the problem of retrieving results from a structured
KB based on a natural language question. KB-QA is an important and challenging
task.

The authors clearly identify the contributions and the novelty of their work,
provide a good overview of the previous work and performance comparison of
their approach to the related methods.

Previous approaches to NN-based KB-QA represent questions and answers as fixed
length vectors, merely as a bag of words, which limits the expressiveness of
the models. And previous work also don’t leverage unsupervised training over
KG, which potentially can help a trained model to generalize.
This paper makes two major innovative points on the Question Answering problem.

1) The backbone of the architecture of the proposed approach is a
cross-attention based neural network, where attention is used for capture
different parts of questions and answer aspects. The cross-attention model
contains two parts, benefiting each other. The A-Q attention part tries to
dynamically capture different aspects of the question, thus leading to
different embedding representations of the question. And the Q-A attention part
also offer different attention weight of the question towards the answer
aspects when computing their Q-A similarity score. 
2) Answer embeddings are not only learnt on the QA task but also modeled using
TransE which allows to integrate more prior knowledge on the KB side. 
Experimental results are obtained on Web questions and the proposed approach
exhibits better behavior than state-of-the-art end-to-end methods. The two
contributions were made particularly clear by ablation experiment. Both the
cross-attention mechanism and global information improve QA performance by
large margins.

The paper contains a lot of contents. The proposed framework is quite
impressive and novel compared with the previous works.

- Weaknesses:
The paper is well-structured, the language is clear and correct. Some minor
typos are provided below.
1. Page 5, column 1, line 421:                                       re-read               
   
 
reread
2. Page 5, column 2, line 454: pairs be    pairs to be

- General Discussion:
In Equation 2: the four aspects of candidate answer aspects share the same W
and b. How about using separate W and b for each aspect? 
I would suggest considering giving a name to your approach instead of "our
approach", something like ANN or CA-LSTM…(yet something different from Table
2).  

In general, I think it is a good idea to capture the different aspects for
question answer similarity, and cross-attention based NN model is a novel
solution for the above task. The experimental results also demonstrate the
effectiveness of the authors’ approach. Although the overall performance is
weaker than SP-based methods or some other integrated systems, I think this
paper is a good attempt in end-to-end KB-QA area and should be encouraged.